# Prostaglandin Transporter and Dipeptidyl Peptidase-4 as New Pharmacological Targets in the Prevention of Acute Kidney Injury in Diabetes: An In Vitro Study

**DOI:** 10.3390/ijms25063345

**Published:** 2024-03-15

**Authors:** Beatriz Gallego-Tamayo, Ángela Santos-Aparicio, Julia Yago-Ibáñez, Laura Muñoz-Moreno, Francisco Javier Lucio-Cazaña, Ana B. Fernández-Martínez

**Affiliations:** 1Universidad de Alcalá, Departamento de Biología de Sistemas, Facultad de Medicina y Ciencias de la Salud, Campus Universitario, Crtra A2, Km. 33,600, 28805 Alcalá de Henares, Spain; bgallegotamayo@gmail.com (B.G.-T.); angela.santosa@uah.es (Á.S.-A.); laura.munozm@uah.es (L.M.-M.); 2Departamento de Biología, Universidad Autónoma de Madrid, 28049 Madrid, Spain; julia.yago@uam.es (J.Y.-I.); anab.fernandez@uam.es (A.B.F.-M.)

**Keywords:** acute kidney injury, cyclo-oxygenase-2, diabetes mellitus, dipeptidyl peptidase-4, prostaglandin E2, prostaglandin transporter, proximal tubular cells, sepsis

## Abstract

The probability of acute kidney injury (AKI) is higher in septic diabetic patients, which is associated with, among other factors, proximal tubular cell (PTC) injury induced by the hypoxic/hyperglycemic/inflammatory microenvironment that surrounds PTCs in these patients. Here, we exposed human PTCs (HK-2 cells) to 1% O_2_/25 mM glucose/inflammatory cytokines with the aim of studying the role of prostaglandin uptake transporter (PGT) and dipeptidyl peptidase-4 (DPP-4, a target of anti-hyperglycemic agents) as pharmacological targets to prevent AKI in septic diabetic patients. Our model reproduced two pathologically relevant mechanisms: (i) pro-inflammatory PTC activation, as demonstrated by the increased secretion of chemokines IL-8 and MCP-1 and the enhanced expression of DPP-4, intercellular leukocyte adhesion molecule-1 and cyclo-oxygenase-2 (COX-2), the latter resulting in a PGT-dependent increase in intracellular prostaglandin E_2_ (iPGE_2_); and (ii) epithelial monolayer injury and the consequent disturbance of paracellular permeability, which was related to cell detachment from collagen IV and the alteration of the cell cytoskeleton. Most of these changes were prevented by the antagonism of PGE_2_ receptors or the inhibition of COX-2, PGT or DPP-4, and further studies suggested that a COX-2/iPGE2/DPP-4 pathway mediates the pathogenic effects of the hypoxic/hyperglycemic/inflammatory conditions on PTCs. Therefore, inhibitors of PGT or DPP-4 ought to undergo testing as a novel therapeutic avenue to prevent proximal tubular damage in diabetic patients at risk of AKI.

## 1. Introduction

Acute kidney injury (AKI), which occurs in approximately 10% of all hospitalized individuals and nearly 50% of those admitted to intensive care units [1], can originate from multiple etiologies, such as coronavirus disease 2019, sepsis, heart or respiratory failure, trauma, ischemia-reperfusion or nephrotoxic drugs. Earlier research has underscored the importance of hypoxia and inflammation in the development of AKI [2,3,4,5] while identifying diabetes as a risk factor for AKI in sepsis [6]. Irrespective of AKI’s etiology, proximal tubular cells (PTCs) have recently been proposed as the pathogenic and therapeutic target in AKI [7], not only because in most forms of AKI they are the passive primary victim of injury but also because they may become activated so that they acquire an inflammatory phenotype that contributes actively to kidney injury [5]. This inflammatory phenotype is characterized by the upregulation of leukocyte adhesion molecules and inflammatory mediators, including (among others) cyclo-oxygenase-2 (COX-2)-derived prostaglandins [8,9], inflammatory cytokines and chemokines [5]. These effectors contribute to renal parenchymal cell damage and to the recruitment of immune cells, thereby aggravating local inflammatory injury through the release of more pro-inflammatory mediators [5].

The involvement of the enzyme COX-2 in the local regulation of inflammatory processes through the release to the extracellular medium of its products is widely recognized. PGE_2_ is the major inflammatory product of COX-2 [10] and as such collaborates in PTC damage [11,12,13,14,15,16,17]. However, in several experimental models [12,15,16,17] we have found that PTC damage is prevented not only by the inhibition of COX-2 or antagonism of EP receptors (PGE_2_ receptors), but also by the inhibition of the prostaglandin transporter PGT, which is responsible for the majority of the cellular uptake of PGE_2_ [18]. Although it is commonly accepted that PGE_2_ re-uptake by PGT leads to the cellular enzymatic catabolism of this *intracellular* PGE_2_ (iPGE_2_), our previous results indicate that iPGE_2_-dependent mechanisms also have a relevant role in mediating PTC injury [12,15,16,17].

Recent studies also suggest that the aminopeptidase dipeptidyl peptidase-4 (DPP-4) may modulate renal inflammation. DPP-4 exists as a cell membrane bound or soluble form in body fluids and, through its exopeptidase action, reduces the biological activity of immunomodulatory proteins, chemokines and peptide hormones. Among the organs, the kidney exhibits the highest DPP-4 expression levels (when adjusted for organ weight), and this is particularly true in the brush border membrane of PTCs [19]. DPP-4 inhibitors, commonly referred to as gliptins, are oral anti-hyperglycemic medications that lower glucose levels by inhibiting the degradation of incretins by DPP-4. This leads to increased insulin secretion and decreased glucagon secretion [20]. Nevertheless, DPP-4 inhibitors have also exhibited a positive effect on several indicators of renal inflammation, regardless of glucose concentrations, in diabetic and non-diabetic renal injuries [20,21,22,23,24].

In summary, both iPGE_2_- and DPP-4-dependent mechanisms could contribute to the inflammatory activation and injury of PTCs that lead to AKI in septic diabetic patients. Here, this hypothesis was confirmed in PTCs that were exposed to hypoxia, hyperglycemia and inflammatory cytokines (i.e., the three main microenvironmental conditions likely affecting PTCs in septic diabetic patients at risk of AKI [25,26]).

## 2. Results

### 2.1. Hypoxic/Hyperglycemic/Inflammatory Conditions Determine Pro-Inflammatory Responses in Human Renal Proximal Tubular HK-2 Cells: Dependency on DPP-4, COX-2, EP Receptors and PGT

We have hypothesized that PTCs under hypoxic/hyperglycemic/inflammatory conditions contribute to AKI through a pro-inflammatory response that involves an increase in their expression of leukocyte adhesion molecules and inflammatory chemokines in a DPP-4-, COX-2-, EP receptor- and PGT-dependent manner (which, in turn, would facilitate acute inflammatory responses involving neutrophils and monocytes/macrophages [4]). As shown in Figure 1a,b, HK-2 cells under these conditions exhibited increased expression of all the pro-inflammatory molecules analyzed, with these results being similar to other studies in several in vitro models related to the role of PTCs in AKI in either diabetic or non-diabetic contexts [27,28,29,30,31]. Pre-treatment with sitagliptin diminished the effects of hypoxic/hyperglycemic/inflammatory conditions on the expression of ICAM-1 and the release of MCP-1, but it had no effect on the release of IL-8 (Figure 1a,b). This suggests that DPP-4 is a relevant factor for the acquisition of some (but not all) inflammatory markers in PTCs in our experimental setting.

Regarding the role of the COX-2/PGE_2_/iPGE_2_ cascade, bromocresol green and celecoxib—respective inhibitors of PGT and COX-2—prevented the changes induced by hypoxic/hyperglycemic/inflammatory conditions (Figure 1c,d). These results indicated that the pro-inflammatory activation of HK-2 cells was mediated by the PGT-dependent re-uptake of prostaglandins produced by HK-2 cells in a COX-2-dependent manner. To verify the specific involvement of PGE_2_, we examined the preventive effect of the EP1-3 receptor antagonist AH6809. As shown in Figure 1c,d, AH6809 prevented the increase in IL-8 and MCP-1 but not the rise in ICAM-1.

### 2.2. Hypoxic/Hyperglycemic/Inflammatory Conditions Induce Epithelial Monolayer Injury and Alteration of Paracellular Permeability: Dependency on DPP-4, COX-2, EP Receptors and PGT

The kidney’s tubular epithelium acts as a barrier, regulating controlled paracellular and transcellular transport of ions and other substances. During AKI, PTC injury leads to cell detachment from the proximal tubule, which contributes to the loss of proximal tubule barrier function as well as to obstruction within the tubules and back-leakage of glomerular filtrate [7,32]. A way to determine the loss of adherent HK-2 cells is based upon the fact that PTCs are connected to the basal membrane through collagen IV (the most abundant component of the basal membrane) and other extracellular matrix proteins [33]. Therefore, we studied the effect of hypoxic/hyperglycemic/inflammatory conditions on cell shedding from collagen IV and found that it increased in a manner sensitive to the DPP-4 inhibitor sitagliptin, the COX-2 inhibitor celecoxib, the EP1-3 receptor antagonist AH6809 and the PGT inhibitor bromocresol green (Figure 2a). These results indicated that cell detachment from collagen IV in our experimental conditions is dependent on the DPP-4 and COX-2/PGE_2_/iPGE_2_ cascade.

HK-2 cells experiencing cell death lose their adherence and are subsequently eliminated from the cell population. Accordingly, cell death leads to a reduction in crystal violet staining within the culture [34]. Therefore, we studied the effect of our experimental conditions on the population of attached cells and found that it was significantly decreased (Figure 2b, upper panel), which was coincident with an increase in cell death as assessed by a trypan blue exclusion test of cell viability (Figure 2b, lower panel). The PGT inhibitor bromocresol green was equally effective in both assays. The EP1-3 receptor antagonist AH6809 fully prevented the loss of attached cells (crystal violet assay) but only partially prevented the loss of cell viability (trypan blue assay). In addition, both the DPP-4 inhibitor sitagliptin and the COX-2 inhibitor celecoxib prevented the loss of attached cells (crystal violet assay) better than the loss of cell viability (trypan blue assay). It is likely that these discrepancies appear because both assays are not equivalent in terms of cell death: while the trypan blue assay is only affected by cell death, the crystal violet assay rather reflects cell count because it is dependent not only on cell death but also on cell proliferation. Therefore, the trypan blue assay is more suitable for the specific quantification of cell death.

PTCs are interconnected through a series of intercellular junctions that play a crucial role in forming and maintaining the proximal tubule barrier. The cytoskeleton, and particularly its actomyosin ring, is a pivotal regulator of the intercellular junctions [35,36]. In order to explore the changes in epithelial cell morphology and monolayer integrity, we used phalloidin staining of the F-actin cytoskeleton. Upon 24 h under hypoxic/hyperglycemic/inflammatory conditions, HK-2 cells underwent a dramatic alteration in their morphological features, which included a remarkable reorganization of the actin cytoskeleton, leading to a significant depletion of structures based on F-actin. This led to the disruption of intercellular contacts at various sites, causing gaps to appear between the cells, ultimately leading to the disruption of the monolayer (Figure 2c), which was prevented by the DPP-4 inhibitor sitagliptin or intervention in the COX-2/PGE_2_/iPGE_2_ pathway (i.e., pre-incubation with the EP1-3 receptor antagonist AH6809, the PGT inhibitor bromocresol green or the COX-2 inhibitor celecoxib).

Ensuring normal paracellular transport in renal tubules is crucial for maintaining kidney functions. The results shown in Figure 2a–c suggest that paracellular permeability of PTCs might increase under hypoxic/hyperglycemic/inflammatory conditions as a consequence of the alterations in PTC morphology and the loss of monolayer integrity. Therefore, we next analyzed in HK-2 cells the effect of our experimental conditions on paracellular permeability. As shown in Figure 2d, exposure of HK-2 cells to hypoxic/hyperglycemic/inflammatory conditions resulted in a notable rise in the apical-to-basolateral transepithelial flux of FITC-labelled dextran (70 kDa) in comparison to cells under control conditions. Pre-incubation with the DPP-4 inhibitor sitagliptin or with inhibitors of the COX-2/PGE_2_/iPGE_2_ pathway avoided the changes induced by the diabetic-like microenvironment.

Taken together, the data shown in Figure 2 suggest (i) that the microenvironment that surrounds PTCs in diabetic patients at risk of AKI may lead to the derangement of the proximal tubule barrier function as a consequence of cytoskeleton-dependent changes in PTC morphology, PTC injury and PTC detachment from the proximal tubule basal membrane and (ii) that DPP-4 activity and the COX-2/PGE_2_/iPGE_2_ pathway play a relevant role in mediating these effects because many of them are prevented by the DPP-4 inhibitor sitagliptin, the COX-2 inhibitor celecoxib, the EP1-3 receptor antagonist AH6809 or the PGT inhibitor bromocresol green.

### 2.3. Human Proximal Tubular HK-2 Cells Exposed to Hypoxic/Hyperglycemic/Inflammatory Conditions Exhibit Increased DPP-4 Expression and Activity: Dependency on COX-2, EP Receptors and PGT

In our experimental setting we found an increase in the expression and activity of DPP-4 (Figure 3a) as well as the expression of COX-2 and iPGE_2_ in HK-2 cells (Figure 3b). Because COX-2 activity might be regulated by DPP-4 [37,38,39,40] (although there are no specific studies in renal tissues), we also assessed the inhibitory effect of pre-incubation with the DPP-4 inhibitor sitagliptin on the increase in iPGE_2_ and COX-2 expression induced by our experimental conditions. Sitagliptin prevented (slightly, but significantly) the stimulatory effect of these conditions on COX-2 expression, but it had no effect on the increase in iPGE_2_ (Figure 3b).

Next, we sought to determine the dependency on COX-2, EP receptors and PGT of the changes found in DPP-4 expression. To this end, cells were pre-incubated with the EP1-3 receptor antagonist AH6809, the PGT inhibitor bromocresol green or the COX-2 inhibitor celecoxib before being exposed to hypoxic/hyperglycemic/inflammatory conditions. Our results (Figure 4a) indicated that these compounds fully prevented the increase in DPP-4 expression. Therefore, EP receptors and the COX-2/PGT-dependent increase in iPGE_2_ mediate the changes in DPP-4 expression and the activity induced by our experimental conditions in HK-2 cells.

In our experimental setting, the expressions of DPP-4 and COX-2 were mutually dependent on each other but to a different degree: while DPP-4 expression was fully dependent on COX-2, EP receptors and PGT (Figure 4a), COX-2 expression was only slightly dependent on DPP-4 (Figure 4b). In order to know more about the interdependency between both enzymes, we analyzed their expression in time course experiments and found that the increase in COX-2 under hypoxic/hyperglycemic/inflammatory conditions is an early event that precedes the augment in DPP-4 expression (Figure 4b). This finding indicates that the increase in DPP-4 expression, which is only evident after 24 h of incubation, is due to the earlier upregulation of COX-2, which is probably independent of DPP-4. This hypothesis was confirmed by the fact that the earlier increase in COX-2 expression was not prevented by the DPP-4 inhibitor sitagliptin (Figure 4c).

## 3. Discussion

The present work is based on the hypothesis that an increase in DPP-4 activity and *intracellular* PGE_2_ content plays a relevant role in the damage to PTCs under hyperglycemic/hypoxic/inflammatory conditions that may lead to AKI in septic diabetic patients. In order to check this hypothesis, we generated an in vitro model of the diabetic/inflammatory milieu in which human renal proximal tubular HK-2 cells were exposed to 1% O_2_, 25 mM glucose and a cytokine cocktail (to emulate systemic inflammation instead of triggering a particular inflammatory pathway). We start here with a discussion of the similarities between proximal tubular alterations in AKI and the results found in our experimental model: During AKI, PTC injury leads to epithelial monolayer injury due to cell detachment from the basal membrane, alteration of the cell cytoskeleton and cell death, altogether contributing to the loss of proximal tubule barrier function [32]. These effects, which were mimicked by HK-2 cells under hypoxic/hyperglycemic/inflammatory conditions (Figure 2), lead to a rapid decline in renal function [41] not only due to the altered tubular permeability but also because detached PTCs cause obstruction of the renal tubules and back-leakage of glomerular ultrafiltrate [7]. In addition to these changes related to PTC injury, our experimental model (Figure 1) also reproduced the pro-inflammatory activation of PTCs that contributes to leukocyte-mediated renal injury in AKI [4]: under hypoxic/hyperglycemic/inflammatory conditions, there was an increase in HK-2 cells in the secretion of the chemokines IL-8 and MCP-1 and the expression of the leukocyte adhesion molecule ICAM-1 (Figure 1). Empirical data indicate that the renal tubules can produce chemokines that serve as signals for attracting leukocytes into the renal interstitium [42]. In addition, PTC expression of adhesion molecules is also required for the firm attachment of leukocytes previously directed to them through chemotaxis [4]. The acquisition by PTCs of these pro-inflammatory properties leads to the recruitment of immune cells, thereby aggravating the damage of PTCs through leukocyte-dependent inflammatory responses [43]. In this context, given its well-established role as an inflammatory mediator, PGE_2_ is highly likely to be involved in the instauration and progression of AKI, as shown by the known activation of the renal inflammatory COX-2/PGE_2_ cascade in experimental models of AKI [9] and our own results. In summary, the changes found in HK-2 cells exposed to hypoxic/hyperglycemic/inflammatory conditions suggest that our experimental model reproduces some of the pathogenic mechanisms that are probably involved in the onset and progression of AKI in septic diabetic patients.

The vast majority of the effects of the hypoxic/hyperglycemic/inflammatory conditions were avoided by the pharmacologic inhibition of COX-2 or PGT or the pharmacologic antagonism of EP receptors. This indicates that the COX-2/PGE_2_/iPGE_2_ axis plays a relevant role in the mentioned effects. In this regard, one could propose the use of non-steroidal anti-inflammatory drugs—a widely used category of COX inhibitor drugs, commonly prescribed to treat various conditions such as fever, headaches, pain and other common ailments—to prevent AKI in septic diabetic patients. However, COX inhibition leads to a drastic reduction in the renal production of PGE_2_ and other vasodilating prostaglandins that act on the renal afferent arterioles [44,45,46]. This ultimately results in effects resembling hypotension, which, in turn, can contribute to the development of AKI, chronic kidney disease and other nephropathies [44], which is a major obstacle in the use of non-steroidal anti-inflammatory drugs or antagonists of EP receptors in the prevention and treatment of AKI in diabetic patients. In this context, targeting PGT (and thereby the intracellular effects of PGE_2_) could potentially serve as a promising and appealing novel therapeutic approach. This view is in good agreement with our earlier findings showing that the inhibition of PGT in several experimental models provides beneficial effects against PTC injury [12,15,16,17].

The DPP-4 inhibitor sitagliptin had basically the same protective effects on HK-2 cells as the inhibitors of COX-2, EP receptors and PGT (Figure 1 and Figure 2). This fact and the results showing that these inhibitors also prevent the increase in DPP-4 under hypoxic/hyperglycemic/inflammatory conditions (Figure 3 and Figure 4) strongly suggest that the effects of these conditions in HK-2 cells are mediated by a COX-2/iPGE_2_/DPP-4 cascade. Accordingly, we propose that DPP-4 mediates the cellular effects of the activation of the COX-2/PGE_2_/iPGE_2_ axis in our experimental setting, which should be confirmed by further specific studies. It is currently unknown whether DPP-4 renal activation contributes to AKI: whereas several studies showed that DPP-4 inhibitors had neutral effects on the risk of AKI in diabetic patients [47,48], extensive evidence indicates that DPP-4 inhibition results in the protection of the kidneys from various types of renal conditions, including AKI [21,49,50,51,52,53,54,55]. Among this evidence, two studies specifically point to the preventive effect of DPP-4 inhibitors against the development of AKI in diabetic patients: first, the administration of DPP-4 inhibitors to patients with type 2 diabetes mellitus has been linked to reduced chances of AKI within 120 days, in comparison to both diabetic and non-diabetic control groups [49]; secondly, in diabetic cancer patients undergoing cisplatin treatment, the incidence of AKI was significantly lower (25% vs. 64%) in those treated with DPP-4 inhibitors [55]. In this context, our current results agree well with the idea that DPP-4 in proximal tubules may play a relevant role in the pathogenic mechanisms responsible for the development of AKI in septic diabetic patients.

In conclusion, taken together, our results support the view that a COX-2/iPGE_2_/DPP-4 cascade might have a relevant role in the proximal tubular damage in septic diabetic patients; therefore, inhibitors of PGT and DPP-4 should be tested in animal models of sepsis as new potential therapeutic approaches to prevent AKI in these patients. Whether these inhibitors also protect PTCs once AKI is established deserves further investigation.

## 4. Materials and Methods

### 4.1. Reagents and Antibodies

Celecoxib, bromocresol green, anti-β-actin antibody, DAPI, anti-mouse IgG and anti-rabbit IgG peroxidase conjugated, human collagen IV, crystal violet, trypan blue, ITS (Insulin (10 mg/L)/Transferrin (5.5 mg/L)/Selenium (5 μg/L)) and fluorescein-labelled 70-kDa dextran were obtained from Sigma Aldrich (St. Louis, MO, USA); the antibodies anti-COX-2 and anti-PGE2 were purchased from Abcam (Cambridge, UK) and anti-ICAM-1 was from Cell Signalling (Danvers, MA, USA); ProLong Gold Antifade Reagent^®^ was acquired from Invitrogen (Waltham, MA, USA); Phalloidin CruzFluorTM 594 conjugated and bovine serum albumin (ChemCruz^®^) were obtained from Santa Cruz (Dallas, TX, USA); interferon-γ (IFN-γ), tumor necrosis factor-α (TNF-α) and interleukins 1α, 1β and 2 (IL-1α, IL-1β and IL-2) were purchased from Peprotech (London, UK). Reverse transcriptase and Supreme qPCR Green Master Mix were obtained from NZYTECH (Lisboa, Portugal). BCA Protein Assay Kit, Trizol reagent and cocktail of proteases inhibitors were purchased from Thermo Fisher (Waltham, MA, USA). AH6809 was acquired from Cayman Chemical (Ann Arbor, MI, USA) and sitagliptin from TEBU-BIO (Barcelona, Spain).

### 4.2. Cell Culture and Experimental Conditions

DMEM/F12 and DMEM high glucose (HG, 25 mM glucose) were provided by ThermoFisher (Grand Island, NY, USA); DMEM low glucose (LG, 5.5 mM glucose) was acquired from Labclinics (Barcelona, Spain). HK-2 cells (human PTC) were obtained from the American Type Culture Collection (ATCC) (Rockville, MD, USA). They were cultured under 5% CO_2_ at 37 °C in DMEM/F12 and supplemented with 10% fetal bovine serum (Gibco, Thermo Fisher (Waltham, MA, USA)) and ITS. Four days prior to commencing the experiments, the cell culture medium was switched to DMEM LG supplemented with 10% fetal bovine serum and ITS.

For all experimental setups, when HK-2 cells reached 70–90% confluence, they were subjected to control conditions (DMEM LG supplemented with 0.5% fetal bovine serum and ITS and 21% O_2_) or to hypoxia/hyperglycemia/inflammatory cytokines (1% O_2_, DMEM HG (with 0.5% FBS) and TNF-α: 50 ng/mL, IFNγ: 20 ng/mL, IL-1α: 10 ng/mL, IL-1β: 10 ng/mL, IL-2: 40 ng/mL). Work under hypoxic conditions was conducted within an In vivo200 hypoxia workstation (Ruskinn Technology, West Yorkshire, UK). Accumulation of hypoxia inducible factor-1α, a hypoxia target, was tested for model verification.

When required, HK-2 cells were pre-treated with specific inhibitors such as sitagliptin (10 µM), AH6809 (10 µM), bromocresol green (50 µM) or celecoxib (2 µM), targeting DPP-4, EP1-3 receptors, PGT or COX-2, respectively. None of these inhibitors were toxic at the concentrations used, as assessed by crystal violet assay.

### 4.3. Protein Extraction and Western Blot Assay

Cells (1.5 × 10^5^ cells/well) were cultured for 24 h in six-well plates and subjected to the aforementioned treatments. Afterwards, they were scraped into lysis buffer, and the whole-cell lysate protein content was measured using a BCA Protein Assay Kit. Subsequently, equal amounts of protein were separated using 8–15% SDS–PAGE and transferred onto a PVDF membrane (Bio-Rad Laboratories, Hercules, CA, USA). After blocking, the membranes were incubated overnight at 4 °C with the primary antibodies. Finally, the membranes were incubated with peroxidase-conjugated secondary antibody, and the immunocomplex signals were visualized using an enhanced chemiluminescence reagent and the ImageQuant LAS 500 System (General Electric Healthcare, Little Chalfont, Buckinghamshire, UK). Band densities were quantified using Fiji software (https://micron.ox.ac.uk (European mirror) accessed on 22 October 2023).

### 4.4. RNA Isolation and Real-Time PCR (q-RT-PCR) Analysis

Total RNAs were isolated from HK-2 cells grown in six-well plates using TRIZOL reagent. Reverse transcription was carried out from one microgram of RNA. q-RT-PCR was conducted using the Supreme qPCR Green Master Mix. The primer sequences for DPP-4 and GAPDH were as follows: DPP-4 sense, 5′-ACGTGAAGCAATGGAGGCAT-3′; Antisense, 5′-GTGACCATGTGACCCACTGT-3′; GAPDH sense, 5’-CAAGGGCATCCTGGGCTAC-3´; Antisense, 5´-GCCCCAGCGTCAAAGGTGGA-3´. Relative differences in DPP-4 expression were calculated using the ΔΔCt method, with GAPDH mRNA serving as the endogenous reference.

### 4.5. DPP-4 Activity

Cells (1.5 × 10^5^ cells/well) were cultured for 24 h in six-well plates and subjected to the aforementioned treatments (see Section 4.2). Afterwards, the assay for DPP-4 activity (Abcam, Cambridge, UK) in the supernatants was performed according to the manufacturer instructions, and fluorescence was determined in a kinetic mode using a fluorescent plate reader (VICTOR X4, Perkin Elmer, Waltham, MA, USA). DPP-4 activity (pg/mL/min) was corrected by protein content, and the results were expressed as relative units vs. control.

### 4.6. MCP-1 and IL-8 Analysis in the Extracellular Medium

Cells (1.5 × 10^5^ cells/well) were cultured for 24 h in six-well plates and subjected to the aforementioned treatments (see Section 4.2). Protein quantitation in cell culture supernatants was performed using commercially available ELISA kits (IL-8 ELISA and MCP-1 ELISA were from Abcam, Cambridge, UK) as per the manufacturer’s instructions. Assays were performed using 0.5 µL and 50 µL of cell culture supernatants, respectively. Absorbance was measured using a microplate reader, FL600 (BioTek, Winooski, VT, USA). The protein level was calculated using the manufacturer´s instructions. Whole-cell lysate protein content was used to normalize the results.

### 4.7. Immunofluorescence Assessment

Cells (1.5 × 10^5^ cells/well) were cultured for 24 h in 12 mm^2^ coverslips (3 × 10^4^ cells/coverslip) and subjected to the aforementioned treatments (see Section 2.2). Subsequently, the cells were first fixed using 4% paraformaldehyde and 0.1% Triton X-10. Then, they were blocked with 4% bovine serum albumin and exposed to phalloidin or anti-PGE2 antibodies. After washing with PBS, cells were incubated with the anti-rabbit secondary antibodies Alexa-Fluor 568 IgG (H-L). To visualize cell nuclei, DAPI staining was carried out. The coverslips were then mounted with ProLong solution and observed using a Leica SP5 confocal microscope (Leica Microsystems, Wetzlar, Germany), operated through the Confocal Microscopy Service (ICTS ‘NANBIOSIS’ U17) of the Biomedical Research Networking Centre on Bioengineering, Biomaterials, and Nanomedicine (CIBER-BBN) at the University of Alcala, Madrid, Spain (http://www.uah.es/enlaces/investigacion.shtm (accessed on 5 September 2023)). The immunofluorescence intensity associated with iPGE2 was quantified following digital capture using Fiji software (https://micron.ox.ac.uk (European mirror) accessed on 22 October 2023).

### 4.8. Crystal Violet and Trypan Blue Assays

Crystal Violet, a dye binding to cellular proteins and DNA, offers insight into adherent cell density. In this procedure, cells (3 × 10^4^ cells/well) were cultured for 24 h in 24-well plates and subjected to the aforementioned treatments (see Section 2.2). Following treatment, cells were exposed to a crystal violet solution (composed of 0.25% crystal violet and 70% aqueous methanol) for 10 min and thoroughly washed with PBS. The dye bound to adherent cells was subsequently extracted using 500 μL of 1% SDS, and the absorbance of the resulting solution was quantified by spectrophotometry at a 570 nm wavelength using a BioTek microplate reader, FL600 (Winooski, VT, USA).

The trypan blue dye exclusion test offers insight into cell death. In this procedure, cells were cultured and treated in the same way as cells for the crystal violet assay. Detached cells from the experimental process were recovered via centrifugation of their culture medium (500× *g* for 5 min). Adherent cells were also recovered after being incubated with trypsin. Both types of cells were resuspended in 0.4% trypan blue in culture medium (1:1 *v*/*v*) and mixed. The viable cells, characterized by a clear cytoplasm, were manually counted using a light microscope and a hemocytometer. Cells with damaged membranes and a blue cytoplasm were considered non-viable, and they were also counted. The percentage of deceased cells was calculated based on this assessment.

### 4.9. Assessment of Cell Detachment

Human collagen IV was used to coat 6-well plates as previously described [17]. HK-2 cells were then cultivated in these plates and subjected to the treatments specified earlier (refer to Section 4.2). Detached cells from the experimental process were recovered via centrifugation of their culture medium. Adherent cells were also recovered after being incubated with trypsin. Both types of cells were resuspended in culture medium and, subsequently, manual cell counting was performed as above. The percentage of floating cells was determined in relation to the combined count of floating and adherent cells.

### 4.10. Paracellular Permeability Assay

HK-2 cells were seeded on 24-well transwell dishes with a pore size of 0.4 µm (Corning Costar, Corning, NY, USA) to establish a tightly packed monolayer over a period of 2 days. Following this, the cells were subjected to the treatments described in Section 4.2. Subsequently, they were washed, incubated for 24 h with fluorescein-labelled 70-kDa dextran (100 µg/mL), and the dextran content in the basolateral compartment was quantified as previously described [17] using a fluorescent plate reader (VICTOR X4, Perkin Elmer, Waltham, MA, USA).

### 4.11. Statistical Analysis

The results are presented as the mean ± SEM. Normal distribution of the data was analyzed by the Shapiro–Wilk test. Data distributed along a normal distribution curve underwent one-way analysis of variance (ANOVA) followed by Bonferroni’s test for multiple comparisons or a two-tailed Student’s t-test analysis when applicable. Data that did not fit to a normal distribution underwent the Kruskal–Wallis test, followed by Dunn’s post hoc test. A significance level of *p* < 0.05 was used. Each experiment was conducted a minimum of three times.

## Figures and Tables

**Figure 1 ijms-25-03345-f001:**
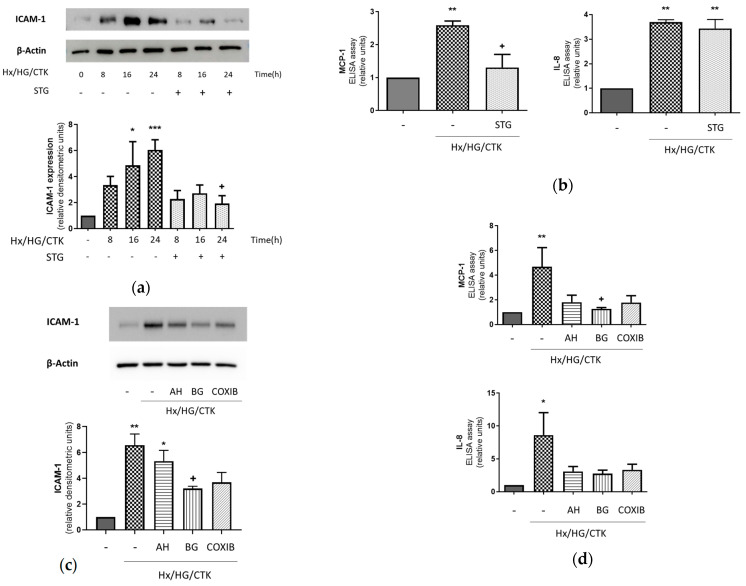
Hypoxic/hyperglycemic/inflammatory conditions determine pro-inflammatory responses in human renal proximal tubular HK-2 cells. Dependency on DPP-4, COX-2, EP receptors and PGT. (**a**) Increase in the expression of leukocyte adhesion molecule ICAM-1 and the effect of the DPP-4 inhibitor sitagliptin. The expression of ICAM-1 was assessed through Western blot analysis. (**b**) Increase in the release of leukocyte chemotactic molecules IL-8 and MCP-1 and the effect of the DPP-4 inhibitor sitagliptin. Release of chemokines to the extracellular medium was determined by ELISA. (**c**,**d**) Prevention of the pro-inflammatory responses by AH6809, bromocresol green and celecoxib. Common information: (1) Cells were exposed for 24 h to control conditions (atmospheric O_2_/5.5 mM glucose) or hypoxic/hyperglycemic/inflammatory conditions (Hx/HG/CTK: 1% O_2_/25 mM glucose/cytokine cocktail of TNF-α: 50 ng/mL; IFN-γ: 20 ng/mL; IL-1α: 10 ng/mL; IL-1β: 10 ng/mL; IL-2: 40 ng/mL). (2) One hour before, the DPP-4 inhibitor sitagliptin (STG, 10 µM), the COX-2 inhibitor celecoxib (COXIB, 2 µM), the EP1-3 receptor antagonist AH6809 (AH, 10 µM) or the PGT inhibitor bromocresol green (BG, 50 µM) was incorporated. (3) Western blot analysis: typical pictures of the results are shown. Data are the densitometric analysis in which ICAM-1 expression was corrected by protein loading (β-actin expression) and represented as relative units over the control. (4) Release of chemokines (ELISA): the protein content of whole-cell lysates was used to normalize the results (absolute values for control MCP-1 and IL-8 were 11.2 ± 2.6pg/µg protein and 3513 ± 717 pg/µg protein, respectively). Controls were incubated for 24 h. (5) Bar charts show the mean ± SEM: * *p* < 0.05 vs. control; ** *p* < 0.01 vs. control; *** *p* < 0.001 vs. control; ^+^ *p* < 0.05 vs. Hx/HG/CTK.

**Figure 2 ijms-25-03345-f002:**
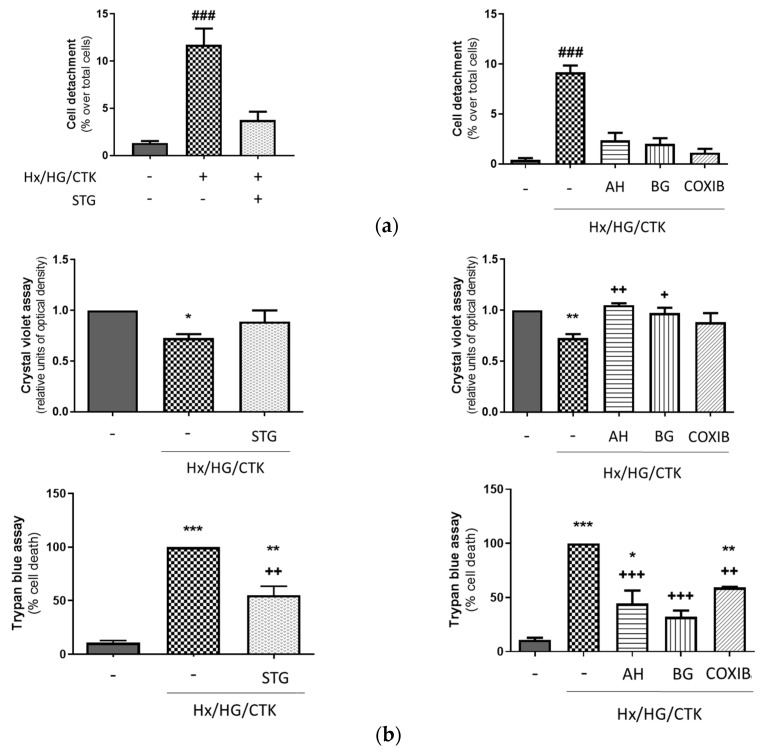
Hypoxic/hyperglycemic/inflammatory conditions induce epithelial monolayer injury and alterations of paracellular permeability. Dependency on DPP-4, COX-2, EP receptors and PGT. (**a**) Detachment from collagen IV. The percentage of floating cells was determined in relation to the combined count of floating and adherent cells. (**b**) Upper panel: assessment of adherent cells. Data from the crystal violet assay were normalized to control values. Lower panel: cell viability. Data from the trypan blue dye exclusion assay were normalized to values found in hypoxic/hyperglycemic/inflammatory conditions. (**c**) Changes in F-actin cytoskeleton. Illustrative microscopy images resulting from phalloidin staining portray the reorganization of the actin cytoskeleton and the compromise of monolayer integrity, as evident from the gaps between cells. (**d**) Increase in paracellular permeability. This was assessed through quantifying the transcellular movement of 70 kDa-FITC dextran across HK-2 cell monolayers cultured on Transwell membranes. Common information: (1) Cells were exposed for 24 h to control conditions (atmospheric O_2_/5.5 mM glucose) or hypoxic/hyperglycemic/inflammatory conditions (Hx/HG/CTK: 1% O_2_/25 mM glucose/cytokine cocktail of TNF-α: 50 ng/mL; IFN-γ: 20 ng/mL; IL-1α: 10 ng/mL; IL-1β: 10 ng/mL; IL-2: 40 ng/mL). (2) One hour before, the DPP-4 inhibitor sitagliptin (STG, 10 µM), the COX-2 inhibitor celecoxib (COXIB, 2 µM), the EP1-3 receptor antagonist AH6809 (AH, 10 µM) or the PGT inhibitor bromocresol green (BG, 50 µM) was incorporated. (3) Bar charts show the mean ± SEM: * *p* < 0.05 vs. control; ** *p* < 0.01 vs. control; *** *p* < 0.001 vs. control; ^+^ *p* < 0.05 vs. Hx/HG/CTK; ^++^ *p* < 0.01 vs. Hx/HG/CTK; ^+++^ *p* < 0.01 vs. Hx/HG/CTK; ^###^ *p* < 0.001 vs. other groups.

**Figure 3 ijms-25-03345-f003:**
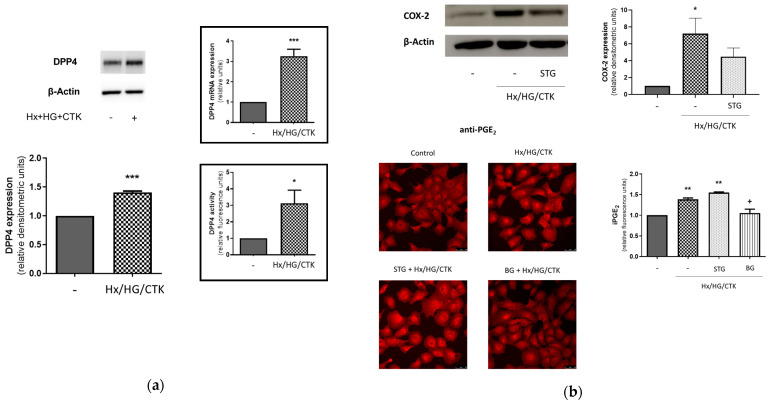
HK-2 cells exposed to hypoxic/hyperglycemic/inflammatory conditions exhibit increased DPP-4 expression and activity and a sitagliptin-dependent increase in COX-2/intracellular PGE_2_. (**a**) Increase in DPP-4 expression and activity. DPP-4 expression was evaluated through Western blot analysis and a q-RT-PCR analysis (inset), and DPP-4 activity was assayed fluorometrically (inset). GAPDH mRNA was used as an endogenous control in the q-RT-PCR analysis, and data from DPP-4 activity were expressed as relative units vs. control and normalized by the protein content in cell lysates (the absolute values for control intracellular and extracellular DPP-4 activity in HK-2 cell supernatant activities were 4159 ± 1163 pmol/min/mg and 16.5 ± 4.3 pmol/min/mL, respectively). (**b**) Increase in COX-2 expression and intracellular PGE_2_ (iPGE_2_) and effect of DPP-4 inhibitor sitagliptin. COX-2 expression (left) was evaluated through Western blot analysis, and iPGE_2_ (right) was determined by immunofluorescence (upper and medium panels: representative microphotographs of the results obtained at original magnification 25×; lower panel: quantitative image analysis by Fiji software) (https://micron.ox.ac.uk (European mirror) (accessed on 22 October 2023)) Common information: (1) Cells were exposed for 24 h to control conditions (C: atmospheric O_2_/5.5 mM glucose) or hypoxic/hyperglycemic/inflammatory conditions (Hx/HG/CTK: 1% O_2_/25 mM glucose/cytokine cocktail of TNF-α: 50 ng/mL; IFN-γ: 20 ng/mL; IL-1α: 10 ng/mL; IL-1β: 10 ng/mL; IL-2: 40 ng/mL). (2) One hour before, the DPP-4 inhibitor sitagliptin (STG, 10 µM) or the PGT inhibitor bromocresol green (BG, 50 µM.) was incorporated (3) Western blot analysis: typical pictures of the results are shown. Data are from the densitometric analysis in which DPP-4 expression or COX-2 expression was corrected by protein loading (β-actin expression) and represented as relative units over the control. (4) Bar charts present the individual experiments and show the mean ± SEM: * *p* < 0.05 vs. control; ** *p* < 0.001 vs. control; *** *p* < 0.001 vs. control; ^+^ *p* < 0.05 vs. Hx/HG/CTK.

**Figure 4 ijms-25-03345-f004:**
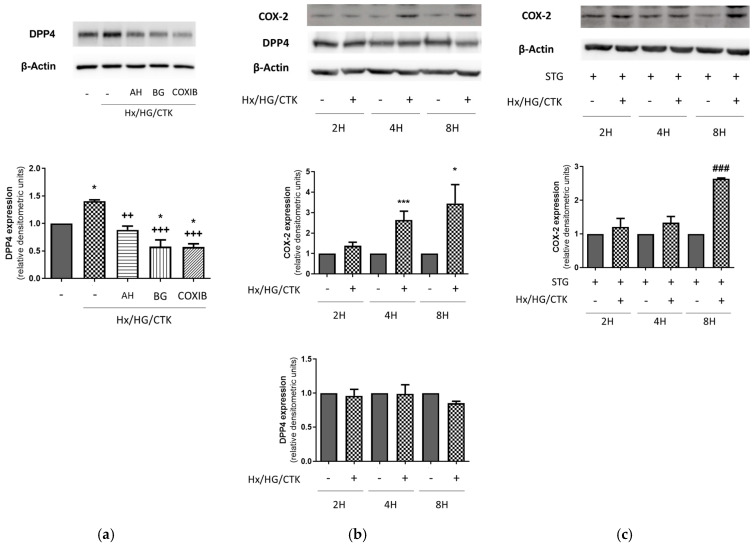
Increased DPP-4 expression in HK-2 cells exposed to hypoxic/hyperglycemic/inflammatory conditions is dependent on COX-2, EP receptors and PGT. (**a**) The prevention by AH6809, bromocresol green and celecoxib of the increase in DPP-4 expression (evaluated through Western blot analysis). (**b**) The increase in COX-2 expression precedes the increase in DPP-4 expression. (**c**) The early increase in COX-2 expression is not avoided by sitagliptin. Common information: (1) Cells were exposed for 24 h to control conditions (C: atmospheric O_2_/5.5 mM glucose) or hypoxic/hyperglycemic/inflammatory conditions (Hx/HG/CTK: 1% O_2_/25 mM glucose/cytokine cocktail of TNF-α: 50 ng/mL; IFN-γ: 20 ng/mL; IL-1α: 10 ng/mL; IL-1β: 10 ng/mL; IL-2: 40 ng/mL). (2) One hour before, the DPP-4 inhibitor sitagliptin (STG, 10 µM), the COX-2 inhibitor celecoxib (COXIB, 2 µM), the EP1-3 receptor antagonist AH6809 (AH, 10 µM) or the PGT inhibitor bromocresol green (BG, 50 µM) was incorporated. (3) Western blot analysis: typical pictures of the results are shown. Data are from the densitometric analysis in which DPP-4 expression or COX-2 expression was corrected by protein loading (β-actin expression) and represented as relative units over the control. (4) Bar charts present the individual experiments and show the mean ± SEM: * *p* < 0.05 vs. control; *** *p* < 0.001 vs. control; ^++^ *p* < 0.01 vs. Hx/HG/CTK; ^+++^ *p* < 0.001 vs. Hx/HG/CTK; ^###^ *p* < 0.001 vs. all other groups.

## Data Availability

Data will be made available on request.

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
