# Peer review of "Prostaglandin Transporter and Dipeptidyl Peptidase-4 as New Pharmacological Targets in the Prevention of Acute Kidney Injury in Diabetes: An In Vitro Study"

_ijms, 2024, doi:10.3390/ijms25063345_

Round 1
Reviewer 1 Report
Comments and Suggestions for Authors
Beatriz et al. submitted the manuscript entitled: Prostaglandin transporter and dipeptidyl peptidase-4 as new pharmacological targets in the prevention of the acute kidney injury in diabetes: an in vitro study, in which they reported their findings on the relation of AKI, PGT and DPP4 pathways. Generally, the authors have extensively investigated on the 2 targets and their efficacy on AKI. I believe this topic will be of interested to the potential readers of IJMS.
I have some suggestions that I’ve broken down into each section.
1. Introduction: a good one, very informative.
2.1:
a) Line 102, here the authors used pretreatment of sitagliptin and observed down regulation of ICAM-1. And the authors mentioned in the manuscript that DPP4 could be upregulated when hypoxic/hyperglycemic/inflammatory conditions was applied to HK2 cells. Here the sitagliptin function as a rescue agent or a therapeutic agent? If sitagliptin was applied after these conditions, will the cell benefit from it?
b) Line 102-104, From the current version, my understanding is upregulation of MCP1 with no difference on IL-8 expression suggested DPP4 as a potential relevant factor? Since IL-8 is also an AKI biomarker, the statement is confusing.
c) The authors tested target ICAM-1/MCP-1/IL-8, which are all AKI targets. Before these targets, hypoxic target like HIF-1α should also be tested for model verification.
2.2:
d) The authors are suggested to explain the potential reason on the result difference of cell viability tests and determine which assay is more suitable for cell death identification in this work.
2.3:
e) The authors are suggested to rearrange figure 3. Separate them into 2 figures if necessary.
Discussion:
f) There are also some references reported that AKI patients cannot benefit from DPP4 inhibition. Although they are against the conclusion of this manuscript, the authors should still cite them
General:
g) Since the authors identify 2 pathways relate to AKI, did the author try to investigate on synergetic effects of these 2 pathways? For example, apply DPP4 inhibitor and COX2 inhibitor simultaneously to HK-2 cells under these conditions.
Reviewer 2 Report
Comments and Suggestions for Authors
B. Gallego-Tamayo and colleagues provided a huge research of mechanisms and possible targets for treatment of AKI in a model of sepsis in diabetes. The manuscript contains strong evidence for the author`s hypothesis based on research applying cutting-edge methods of molecular biology. Some minor changes can be made to polish the manuscript for publication.
- The aim of the study is better to be worded in a common way (Line 76–82). The summary of the results and discussion of significance of the study in the Introduction section looks excessive (Lines 82–88).
- The indication at side effects of COX2 inhibitors in the Abstract section (Line 28) looks also excessive because this was not studied in the current research and was not supported by the results.
- The authors used parametrical tests while they did not check if the distribution was normal (Section 4.11, Line 497–500). In case of non-normally distributed data, non-parametrical tests are more suitable.
- Typos must be corrected (e.g., ‘thee’ at Line 407).
Round 2
Reviewer 1 Report
Comments and Suggestions for Authors
The authors have well addressed on all the issues raised by reviewers.